# Only Pay for What Is Uncertain: Variance-Adaptive Thompson Sampling

**Aadirupa Saha**[1][*] **Branislav Kveton**[2]
[1]Apple, [2]Amazon

## Abstract

Most bandit algorithms assume that the reward variances or their upper bounds are known, and that they are the same for all arms. This naturally leads to suboptimal performance and higher regret due to variance overestimation. On the other hand, underestimated reward variances may lead to linear regret due to committing early to a suboptimal arm. This motivated prior works on variance-adaptive frequentist algorithms, which have strong instance-dependent regret bounds but cannot incorporate prior knowledge on reward variances. We lay foundations for the Bayesian setting, which incorporates prior knowledge. This results in lower regret in practice, since the prior is used in the algorithm design, and also improved regret guarantees. Specifically, we study Gaussian bandits with *unknown heterogeneous reward variances* and develop a Thompson sampling algorithm with prior-dependent Bayes regret bounds. We achieve lower regret with lower reward variances and more informative priors on them, which is precisely why we pay only for what is uncertain. This is the first such result in the bandit literature. Finally, we corroborate our theory with experiments, which demonstrate the benefit of our variance-adaptive Bayesian algorithm over prior frequentist works. We also show that our approach is robust to model misspecification and can be applied with estimated priors.

## 1 Introduction

A *stochastic bandit* (Lai & Robbins, 1985; Auer et al., 2002; Lattimore & Szepesvári, 2020) is an online learning problem where a *learning agent* sequentially interacts with an environment over $n$ rounds. In each round, the agent pulls an *arm* and receives a *stochastic reward*. The mean rewards of the arms are initially unknown and the agent learns them by pulling the arms. Therefore, the agent faces an *exploration-exploitation dilemma*: *explore*, and learn more about the arms by pulling them; or *exploit*, and commit to the arm with the highest estimated reward. An example of this setting is a recommender system, where the arm is a recommendation and the reward is a click.

Most bandit algorithms assume that the reward variance or its upper bound is known. For instance, the confidence intervals in UCB1 (Auer et al., 2002) are derived under the assumption that the rewards are $[0, 1]$, and hence $\sigma^2$-sub-Gaussian for $\sigma = 0.5$. In Bernoulli KL-UCB (Garivier & Cappe, 2011) and Thompson sampling (TS) (Agrawal & Goyal, 2012), tighter confidence intervals are derived for Bernoulli rewards. Specifically, a Bernoulli random variable with either a low or high mean also has a low variance. In general, the reward variance may be hard to specify (Audibert et al., 2009b). *While overestimating it is typically safe, this decreases the learning rate of the bandit algorithm and thus increases its regret. On the other hand, when the variance is underestimated, this may lead to linear regret because the algorithm can commit to an arm without sufficient evidence.*

We motivate learning of reward variances by the following example. Take a movie recommender that learns to recommend highest rated movies in a "Trending Now" carousel. The movies are rated on scale $[1, 5]$. Some movies, such as The Godfather, are classics. Therefore, their ratings are high on average and have low variance. On the other hand, ratings of low-budget movies are often low on average and have low variance, due to the quality of the presentation. Finally, most movies are made for a specific audience, such as Star Wars, and thus have a high variance in ratings. Clearly, any

---

[*]Corresponding email: `aadirupa.saha@gmail.com`

sensible learning algorithm would require fewer queries to estimate the mean ratings of movies with low variances. Since the variance is unknown a priori, adaptation is necessary. This would reduce the overall query complexity and improve statistical efficiency–as one should–because we only pay for what is uncertain. This example is not limited to movies and applies to other domains, such as online shopping. Our work answers the following questions in affirmative:

*Can we quickly learn the right representation of the reward distribution for efficient learning? What is the right dependence of the learner's performance (regret) versus the prior parameters and reward variances? Can we design an algorithm to achieve that rate? Does the regret decrease with lower reward variances and more informative priors on them?*

Unknown reward variances are a major concern and thus have been studied extensively. In the cumulative regret setting, Audibert et al. (2009b) proposed an algorithm based on upper confidence bounds (UCBs) and Mukherjee et al. (2018) proposed an elimination algorithm. In best-arm identification (BAI) (Audibert et al., 2009b; Bubeck et al., 2009), several papers studied the fixed-budget (Gabillon et al., 2011; Faella et al., 2020; Saha et al., 2020; Lalitha et al., 2023) and fixed-confidence (Lu et al., 2021; Zhou & Tian, 2022; Jourdan et al., 2022) settings with unknown reward variances. All above works studied frequentist algorithms. On the other hand, Bayesian algorithms based on posterior sampling (Thompson, 1933; Chapelle & Li, 2011; Agrawal & Goyal, 2012; Russo & Van Roy, 2014; Russo et al., 2018; Kveton et al., 2021; Hong et al., 2022b) perform well in practice, but learning of reward variances in these algorithms is understudied.

We consider the Bayesian setting (Russo & Van Roy, 2014) and introduce it in detail in Section 3. This is because Bayesian algorithms are very practical (Chapelle & Li, 2011; Russo et al., 2018; Kveton et al., 2021) and Bayesian analyses are the only bandit analyses that can capture the dependence on prior (Russo & Van Roy, 2016; Lu & Van Roy, 2019; Hong et al., 2022b;a). More specifically, as the prior becomes more informative, Bayes regret bounds go to zero, and so does the Bayes regret of Thompson sampling. Frequentist regret bounds do not have this behavior because they are proved for any bandit instance, which is unrelated to the prior in the bandit algorithm (Agrawal & Goyal, 2012; 2013b). In fact, all frequentist regret bounds for Bayesian algorithms assume a sufficiently-wide prior, which is analogous to being uninformative. Taking an expectation of frequentist regret bounds over instances sampled from the prior does not yield the right dependence on the prior. In our experiments, we show that as the prior becomes more informative, our bounds become tighter than the frequentist bounds. This shows the benefit of Bayesian analyses.

Bayesian analyses have two shortcomings. First, they are on average over bandit instances sampled from the prior. This relates the bandit instances to the prior in the bandit algorithm and allows a prior-dependent analysis. Second, to derive closed-form posteriors and use them in the analysis, modeling assumptions are needed. In our work, we assume Gaussian noise, which is less general than the sub-Gaussian noise that is typically used in frequentist analyses (Abbasi-Yadkori et al., 2011). Our contributions are summarized next.

**Contributions:** **(1)** To warm up, we start with Thompson sampling in a $K$-armed Gaussian bandit with *known heterogeneous reward variances* (Section 4). Its regret bound (Theorem 1) decreases as reward variances decrease. It also approaches zero as the prior variances of mean arm rewards go to zero, since a Bayesian learning agent knows the bandit instance with certainty in this case. **(2)** We propose a Thompson sampling algorithm `VarTS` for a $K$-armed Gaussian bandit with *unknown heterogeneous reward variances* (Section 5). `VarTS` maintains a joint Gaussian-Gamma posterior for the mean and precision of the rewards of all arms and samples from it in each round. **(3)** We prove a Bayes regret bound for `VarTS` (Theorem 2), which decreases with lower reward variances and more informative priors on them. This is the first such regret bound. The novelty in our analysis is in handling random confidence interval widths due to random reward variances. The bound captures the same trade-offs as if the variance was known, replaced by the corresponding prior-dependent quantities. **(4)** We evaluate `VarTS` on various types of reward distributions, from Bernoulli to beta to Gaussian (Section 6). Our evaluation shows that `VarTS` outperforms all existing baselines, even with an estimated prior. This showcases the generality and robustness of our method.

## 2 RELATED WORK

Classic $K$-armed bandits have been studied for over three decades (Lai & Robbins, 1985; Lai, 1987). Two popular techniques for solving these problems are UCBs (Auer, 2002; Audibert et al., 2009a;

Ménard & Garivier, 2017) and Thompson sampling (Thompson, 1933; Agrawal & Goyal, 2012; Bubeck & Liu, 2013). Recent works on TS matched the minimax optimal rate in $K$-armed bandits (Jin et al., 2021). Jin et al. (2023) designed a minimax and asymptotically optimal TS. Improved TS algorithms for combinatorial bandits and semi-bandits have also been proposed (Perrault et al., 2020; Perrault, 2022). The focus of our work is on TS that adapts to unknown reward variances.

The beginnings of variance-adaptive algorithms can be traced to Auer et al. (2002). Auer et al. (2002) proposed a variance-adaptive `UCB1` for Gaussian bandits, called `UCB1-Normal`, where the reward distribution of arm $i$ is $\mathcal{N}(\mu_i, \sigma_i^2)$ and $\sigma_i > 0$ is known by the learning agent. The $n$-round regret of this algorithm is $O\left(\sum_{i:\mu_i < \mu_{a^*}} \frac{\sigma_i^2}{\Delta_i} \log n\right)$, where $\Delta_i = \mu_{a^*} - \mu_i$ is the gap of arm $i$ and $a^*$ is the arm with the highest mean reward $\mu_i$. The first UCB algorithm for unknown reward variances with an analysis was `UCB-V` by Audibert et al. (2009b). The key idea in the algorithm is to design high-probability confidence intervals based on empirical Bernstein bounds. The $n$-round regret of `UCB-V` is $O\left(\sum_{i:\mu_i < \mu_{a^*}} \left(\frac{\sigma_i^2}{\Delta_i} + b\right) \log n\right)$, where $b$ is an upper bound on the absolute value of rewards. In summary, variance adaptation in `UCB-V` incurs only a small penalty of $O(bK \log n)$. Mukherjee et al. (2018) proposed an elimination-based variant of `UCB-V` that attains the optimal gap-free regret of $O(\sqrt{Kn})$. While empirical Bernstein bounds are general, they tend to be conservative in practice. This was observed by Garivier & Cappe (2011) and we observe the same trend in our experiments (Section 6). Our work can be viewed as a similar development to `UCB-V` in Thompson sampling. We show that Thompson sampling with unknown reward variances (Section 5) incurs only a slightly higher regret than the one with known variances (Section 4), by a multiplicative factor. Compared to `UCB-V`, the algorithm is highly practical.

Two closest papers to our work are Honda & Takemura (2014); Zhu & Tan (2020). Both papers propose variance-adaptive Thompson sampling and bound its regret. There are three key differences from our work. First, the algorithms of Honda & Takemura (2014); Zhu & Tan (2020) are designed for the frequentist setting. Specifically, they have a fixed sufficiently-wide prior and enjoy a per-instance regret bound under this prior. While this is a strong guarantee, the algorithms can perform poorly when priors are narrower and thus more informative. Truly Bayesian algorithm designs, as proposed in our work, can be analyzed for any informative prior. Second, the analyses of Honda & Takemura (2014); Zhu & Tan (2020) are frequentist. Therefore, they cannot justify the use of more informative priors. We prove regret bounds that decrease with lower reward variances and more informative priors on them. Finally, the regret bounds of Honda & Takemura (2014); Zhu & Tan (2020) are asymptotic. We provide strong finite-time guarantees. We discuss these difference in more detail after Theorem 2 and demonstrate them empirically in Section 6.

Another line of related works are variance-dependent regret bounds for $d$-dimensional linear contextual bandits (Kim et al., 2022; Zhao et al., 2022; 2023; Zhang et al., 2021). These works address the problem of time-dependent variance adaptivity. They derive frequentist regret bounds that scale as $\tilde{O}(\mathrm{poly}(d)\sqrt{1 + \sum_{t=1}^n \sigma_t^2})$, where $\sigma_t^2$ is an unknown reward variance in round $t$. This setting is different from ours in two aspects. First, the reward variances change over time but are fixed across the arms. We do the opposite in our work. Second, their algorithm designs and analyses are frequentist, and thus do not exploit prior knowledge. On the other hand, we focus only on $K$-armed bandits, which is a special case of linear bandits.

## 3 PROBLEM SETUP

**Notation.** The set $\{1, \dots, n\}$ is denoted by $[n]$. The indicator $\mathbb{1}\{E\}$ denotes that event $E$ occurs. We use boldface letters to denote vectors. For any vector $\mathbf{v} \in \mathbb{R}^d$, we denote its $i$-th entry by $v_i$ or $v(i)$. We denote the entry-wise square of $\mathbf{v}$ by $\mathbf{v}^2$. A diagonal matrix with entries $\mathbf{v}$ is $\mathrm{diag}(\mathbf{v})$. $\tilde{O}$ is the big O notation up to polylogarithmic factors. Gaussian, Gamma, and Gaussian-Gamma distributions are denoted by $\mathcal{N}$, $\mathrm{Gam}$, and $NG$, respectively. For any random variables $X$ and $Y$, we abbreviate $\mathbb{E}\left[\mathbb{E}\left[\cdot \mid X, Y\right] \mid X\right]$ as $\mathbb{E}\left[\mathbb{E}\left[\cdot \mid Y\right] \mid X\right]$.

**Setting.** A bandit *instance* is a pair of mean arm rewards and reward variances, $(\boldsymbol{\mu}, \boldsymbol{\sigma}^2)$, where $\boldsymbol{\mu} \in \mathbb{R}^K$ is a vector of mean arm rewards, $\boldsymbol{\sigma}^2 \in \mathbb{R}_{\geq 0}^K$ is a vector of reward variances, and $K$ is the number of arms. We propose algorithms and analyze them for both when the reward variances $\boldsymbol{\sigma}^2$ are known (Section 4) and unknown (Section 5).

**Feedback model.** The agent interacts with the bandit instance $(\boldsymbol{\mu}, \boldsymbol{\sigma}^2)$ for $n$ rounds. In round $t \in [n]$, it pulls an arm and observes its stochastic reward. We denote the pulled arm in round $t$ by $A_t \in [K]$, a stochastic reward vector of all arms in round $t$ by $\boldsymbol{x}_t \in \mathbb{R}^K$, and the reward of arm $i \in [K]$ by $x_{t,i} \in \mathbb{R}$. The rewards are sampled from a Gaussian distribution, $x_{t,i} \sim \mathcal{N}(\mu_i, \sigma_i^2)$. The interactions of the agent up to round $t$ are summarized by a *history* $H_t = (A_1, x_{1,A_1}, \ldots, A_{t-1}, x_{t-1,A_{t-1}})$.

**Bayesian bandit setting.** We consider a *Bayesian multi-armed bandit* (Russo & Van Roy, 2014; Russo et al., 2018; Kveton et al., 2021; Hong et al., 2022b) where the bandit instance is either fully or partially random:

**(i).** When *reward variances are known* (Section 4), the bandit instance $(\boldsymbol{\mu}, \boldsymbol{\sigma})$ is generated as follows. The mean arm rewards are sampled from a Gaussian distribution, $\boldsymbol{\mu} \sim P_0 = \mathcal{N}(\boldsymbol{\mu}_0, \mathrm{diag}(\boldsymbol{\sigma}_0^2))$, where $\boldsymbol{\mu}_0 \in \mathbb{R}^K$ and $\boldsymbol{\sigma}_0^2 \in \mathbb{R}_{\geq 0}^K$ are the prior means and variances of $\boldsymbol{\mu}$, respectively. Both $\boldsymbol{\mu}_0$ and $\boldsymbol{\sigma}_0^2$ are assumed to be known by the agent. The reward variances $\boldsymbol{\sigma}^2$ are also known.

**(ii).** When *reward variances are unknown* (Section 5), the bandit instance $(\boldsymbol{\mu}, \boldsymbol{\sigma})$ is sampled from a Gaussian-Gamma prior. Specifically, for any arm $i$, the mean and variance of its rewards are sampled as $(\mu_i, \sigma_i^{-2}) \sim NG(\mu_{0,i}, \kappa_{0,i}, \alpha_{0,i}, \beta_{0,i})$, where $(\boldsymbol{\mu}_0, \boldsymbol{\kappa}_0, \boldsymbol{\alpha}_0, \boldsymbol{\beta}_0)$ are known prior parameters. This can also be seen as sampling $\sigma_i^{-2} \sim \mathrm{Gam}(\alpha_{0,i}, \beta_{0,i})$ and then $\mu_i \sim \mathcal{N}(\mu_{0,i}, \frac{\sigma_i^2}{\kappa_{0,i}})$. This equivalence follows from the basic properties of the Gaussian-Gamma distribution (Lemma 3 in Appendix).

**Regret.** We measure the $n$-round *Bayes regret* of a learning agent with instance prior $P_0$ as:

$$R_n = \mathbb{E}\left[\sum_{t=1}^n \mu_{A^*} - \mu_{A_t}\right], \tag{1}$$

where $A^* = \mathrm{argmax}_{i \in [K]} \mu_i$ is the *optimal arm*. The above expectation is over mean arm rewards $\boldsymbol{\mu}$ drawn from the prior distribution, unlike in the frequentist setting where $\boldsymbol{\mu}$ would be unknown but fixed (Lattimore & Szepesvári, 2020). The randomness in the expectation includes how the agent chooses $A_t$ and the randomness in bandit feedback $x_{t,A_t} \sim \mathcal{N}(\mu_{A_t}, \sigma_{A_t}^2)$.

We depart from the classic bandit setting (Auer et al., 2002; Abbasi-Yadkori et al., 2011; Lattimore & Szepesvári, 2020) in two major ways. First, we consider Gaussian reward noise, as opposing to a more general sub-Gaussian noise. The Gaussian noise and corresponding conjugate priors lead to closed-form posteriors in our algorithms and analyses, which simplifies them. This is why this choice has been popular in recent Bayesian analyses (Lu & Van Roy, 2019; Kveton et al., 2021; Wan et al., 2021; Hong et al., 2022b;a). Second, our regret is Bayesian, on average over bandit instances. An alternative would be the frequentist regret, which holds for any bounded bandit instance. We choose the Bayes regret because it can capture the relation between the bandit instance and its prior, and thus show benefits of informative priors. We discuss this in depth throughout the paper, and especially after Theorem 1 and Theorem 2. To alleviate concerns about Gaussian posteriors in the algorithm design, we experiment with non-Gaussian bandit problems in Section 6.

# 4 GAUSSIAN BANDIT WITH KNOWN VARIANCES

We start with the Bayesian setting with Gaussian rewards and known heterogeneous reward variances. In Section 4.1, we introduce a Thompson sampling algorithm (Thompson, 1933; Chapelle & Li, 2011; Agrawal & Goyal, 2012) for this setting. Gaussian TS is straightforward and appeared in many prior works, starting with Agrawal & Goyal (2013a). We state and discuss its regret bound in Section 4.2. The bound scales roughly as:

$$\sqrt{n \log n}\sqrt{\sum_{i=1}^K \sigma_i^2 \log\left(1 + n\frac{\sigma_{0,i}^2}{\sigma_i^2}\right)}. \tag{2}$$

One notable property of the bound is that it goes to zero when the reward variances $\sigma_i^2$ or the prior variances of the mean arm rewards $\sigma_{0,i}^2$ do. Although the bound is novel, its proof mostly follows Kveton et al. (2021). We mainly state it to contrast it with the main result in Section 5.

## 4.1 GAUSSIAN THOMPSON SAMPLING

The key idea in our algorithm is to maintain a posterior distribution over the unknown mean arm rewards $\boldsymbol{\mu}$ and act optimistically with respect to samples from it. Since $\boldsymbol{\mu}$ and its rewards are sampled

from Gaussian distributions, the posterior is also Gaussian. Specifically, the posterior distribution of arm $i$ in round $t$ is $\mathcal{N}(\hat{\mu}_{t,i}, \sigma^2_{t,i})$, where $\hat{\mu}_{t,i}$ and $\sigma^2_{t,i}$ are the posterior mean and variance, respectively, of arm $i$ in round $t$. These quantities are initialized as $\hat{\mu}_{1,i} := \mu_{0,i}$ and $\sigma_{1,i} := \sigma_{0,i}$.

Our algorithm is presented in Algorithm 1 (Appendix A) and we call it *Gaussian TS* due to Gaussian rewards. The algorithm works as follows. In round $t$, it samples the mean reward of each arm $i$ from its posterior, $\tilde{\mu}_{t,i} \sim \mathcal{N}(\hat{\mu}_{t,i}, \sigma^2_{t,i})$. After that, the arm with the highest posterior-sampled mean reward is pulled, $A_t := \arg\max_{i \in [K]} \tilde{\mu}_{t,i}$. Finally, the algorithm observes a stochastic reward of arm $A_t$, $x_{t,A_t} \sim \mathcal{N}(\mu_{A_t}, \sigma^2_{A_t})$, and updates its posteriors (Lemma 7 in Appendix) as:

$$\sigma^2_{t+1,i} := \frac{1}{\sigma^{-2}_{0,i} + N_{t+1}(i)\sigma^{-2}_i}, \quad \hat{\mu}_{t+1,i} := \sigma^2_{t+1,i}\left(\frac{\mu_{0,i}}{\sigma^2_{0,i}} + \frac{N_{t+1}(i)\bar{x}_{t+1,i}}{\sigma^2_i}\right),$$

where $\bar{x}_{t+1,i} := \frac{1}{N_{t+1}(i)}\sum_{s=1}^{t}\mathbb{1}\{A_s = i\}x_{s,i}$ is the empirical mean reward of arm $i$ at the beginning of round $t+1$ and $N_{t+1}(i)$ is the number of its pulls. The complete pseudocode of Algorithm 1 is given in Appendix A.

## 4.2 REGRET ANALYSIS

Before analyzing Algorithm 1, we recall the setting again. The mean arm rewards are sampled from a Gaussian prior, $\boldsymbol{\mu} \sim P_0 = \mathcal{N}(\boldsymbol{\mu}_0, \mathrm{diag}(\boldsymbol{\sigma}^2_0))$, where $\boldsymbol{\mu}_0 \in \mathbb{R}^K$ and $\boldsymbol{\sigma}^2_0 \in \mathbb{R}^K_{\geq 0}$ are the prior means and variances of $\boldsymbol{\mu}$, respectively. The reward of arm $i$ in round $t$ is sampled as $x_{t,i} \sim \mathcal{N}(\mu_i, \sigma^2_i)$. Both $\boldsymbol{\mu}_0$ and $\boldsymbol{\sigma}^2_0$, and reward variances $\boldsymbol{\sigma}^2$, are fixed and known. Our regret bound is stated below.

**Theorem 1** (Variance-dependent regret bound for known variances). *Consider the above setting. Then for any $\delta > 0$, the Bayes regret of Gaussian TS is bounded as:*

$$R_n \leq \sum_{i=1}^{K}\sqrt{\frac{2\sigma^2_{0,i}}{\pi}}n\delta + \sqrt{2n}\sqrt{\sum_{i=1}^{K}\sigma^2_i\left(\log(1 + n\sigma^2_{0,i}\sigma^{-2}_i) + \sigma^2_{0,i}\sigma^{-2}_i\right)\log(1/\delta)}.$$

The complete proof of Theorem 1 is in Appendix B. We discuss the bound below.

**Dependence on all parameters of interest and prior.** For $\delta = 1/n$, the bound in Theorem 1 scales roughly as $\tilde{O}\left(\sqrt{n\sum_{i=1}^{K}\sigma^2_i\log\left(1 + n\sigma^2_{0,i}\sigma^{-2}_i\right)} + \sqrt{n\sum_{i=1}^{K}\sigma^2_{0,i}}\right)$. Note that we ignore the first term in Theorem 1, which is order-wise dominated by the second term when $\delta = 1/n$. Our bound has several properties that we discuss next. First, it matches the usual $\sqrt{n}$ dependence of all classic Bayes regret bounds (Russo & Van Roy, 2014; 2016; Lu & Van Roy, 2019). Second, it increases with variances $\sigma^2_i$ of individual arm rewards, which is expected because higher reward variances make learning harder. Third, the bound can be viewed as a generalization of existing bounds that assume homogeneous reward variances. Specifically, Kveton et al. (2021) proved a $\tilde{O}(\sqrt{\sigma^2 K n})$ Bayes regret bound in Lemma 4 under the assumption that the reward distribution of arm $i$ is $\mathcal{N}(\mu_i, \sigma^2)$. We match it when $\sigma_i = \sigma$ for all arms $i$. Fourth, the bound approaches zero as $\sigma_{0,i} \to 0$. In this setting, Gaussian TS knows the mean arm rewards $\mu_i$ almost with certainty because their prior variances $\sigma^2_{0,i}$ are low, and thus no exploration is necessary. This is a unique property of Bayes regret bounds that is not captured by any frequentist analysis, such as that of UCB1-Normal (Auer et al., 2002).

**Regret optimality.** Starting with the seminal works of Russo & Van Roy (2014; 2016), most Bayes regret bounds are $\tilde{O}(\sqrt{n})$ and do not have finite-time instance-dependent lower bounds. Lattimore & Szepesvári (2020) derived a $\Omega(\sqrt{Kn})$ asymptotic lower bound for a $K$-armed bandit as $n \to \infty$ (Theorem 35.1). Our regret bound (Theorem 1) matches this rate when all variances are the same, $\sigma_{0,i} = \sigma_i = 1$ for all $i \in [K]$. In addition, it provides an improved dependence on lower reward variances and more informative priors, which implies faster learning rates in these regimes. In fact, when the prior variances of all mean arm rewards go to zero, $\sigma_{0,i} \to 0$ for all $i \in [K]$, our bound goes to zero; as expected. Therefore, we conjecture that our regret bound is worst-case optimal. The only other lower bound that we are aware of is $\Omega(\log^2 n)$ for a $K$-armed bandit (Theorem 3 in Lai (1987)). This lower bound is asymptotic and applies only to exponential-family reward distributions with a single parameter, which excludes Gaussian distributions because they have two parameters. The

lower bound was recently matched in a Bernoulli bandit by Atsidakou et al. (2023). This indicates that logarithmic upper bounds may be possible in our setting. To conclude, we believe that deriving a tight finite-time $\Omega(\sqrt{Kn})$ lower bound for our setting or a logarithmic upper bound are important problems, and we leave them for future work.

# 5 GAUSSIAN BANDIT WITH UNKNOWN VARIANCES

Our main contribution is the Bayesian setting with Gaussian rewards and unknown heterogeneous reward variances. Similarly to Section 4, we propose a Thompson sampling algorithm for this setting in Section 5.1. We state and discuss its regret bound in Section 5.2. The bound scales roughly as:

$$\sqrt{n \log n}\sqrt{\sum_{i=1}^{K} \frac{\beta_{0,i}}{\alpha_{0,i}-1} \log\left(1 + \frac{n}{\kappa_{0,i}}\right)},$$

where $\frac{\beta_{0,i}}{\alpha_{0,i}-1}$ is a proxy for the reward variance $\sigma_i^2$ in equation 2 and $\kappa_{0,i}^{-1}$ plays the role of $\sigma_{0,i}^2/\sigma_i^2$. Since the dependencies are analogous, the bound captures the structure of the problem similarly to equation 2. *Our main novelty lies in handling the uncertainty of reward variances $\boldsymbol{\sigma}^2$, which is unique among all existing TS proofs.*

## 5.1 ALGORITHM VarTS

As in Algorithm 1, we maintain a posterior distribution over the unknown mean arm rewards $\boldsymbol{\mu}$ and act optimistically with respect to samples from it. The challenge is that the reward variances $\boldsymbol{\sigma}^2$ are also unknown. To overcome it, we use the fact that the posterior of $(\mu_i, \sigma_i^{-2})$ is a Gaussian-Gamma distribution when the prior is and the rewards are Gaussian. We represent the posterior hierarchically, in an equivalent form (Lemma 3 in Appendix), as follows. The posterior distribution of the mean reward of arm $i$ in round $t$ is $\mathcal{N}(\hat{\mu}_{t,i}, \sigma_{t,i}^2)$, where $\hat{\mu}_{t,i}$ and $\sigma_{t,i}^2$ are the posterior mean and sampled variance, respectively. The variance is $\sigma_{t,i}^2 = \frac{1}{\kappa_{t,i}\lambda_{t,i}}$, where $\kappa_{t,i} = O(N_t(i))$ and $\lambda_{t,i}$ is a posterior-sampled reward precision of arm $i$ in round $t$. The posterior distribution of $\lambda_{t,i}$ is $\text{Gam}(\alpha_{t,i}, \beta_{t,i})$, where $\alpha_{t,i}$ and $\beta_{t,i}$ denote its shape and rate parameters, respectively. All posterior parameters are initialized by their prior values $(\mu_{0,i}, \kappa_{0,i}, \alpha_{0,i}, \beta_{0,i})$.

Our algorithm is presented in Algorithm 2 and we call it VarTS, because it adapts to the unknown reward variances of arms. The algorithm works as follows. In round $t$, it first samples the precision of each arm from its posterior, $\lambda_{t,i} \sim \text{Gam}(\alpha_{t,i}, \beta_{t,i})$, and then it samples the mean arm reward from its posterior, $\tilde{\mu}_{t,i} \sim \mathcal{N}(\hat{\mu}_{t,i}, \frac{1}{\kappa_{t,i}\lambda_{t,i}})$. After that, the arm with the highest posterior-sampled mean reward is pulled, $A_t := \arg\max_{i \in [K]} \tilde{\mu}_{t,i}$. Finally, the algorithm observes a stochastic reward of arm $A_t$, $x_{t,A_t} \sim \mathcal{N}(\mu_{A_t}, \sigma_{A_t}^2)$, and updates its posteriors (lines 7–13 in Algorithm 2). The complete pseudocode Algorithm 2 is given in Appendix C.

## 5.2 REGRET ANALYSIS

We recall that the bandit instance $(\boldsymbol{\mu}, \boldsymbol{\sigma})$ is sampled from a Gaussian-Gamma distribution: for any arm $i$, the mean and variance of its rewards are sampled as $(\mu_i, \sigma_i^{-2}) \sim NG(\mu_{0,i}, \kappa_{0,i}, \alpha_{0,i}, \beta_{0,i})$, where $(\boldsymbol{\mu}_0, \boldsymbol{\kappa}_0, \boldsymbol{\alpha}_0, \boldsymbol{\beta}_0)$ are known prior parameters. This can also be seen as first sampling $\sigma_i^{-2} \sim \text{Gam}(\alpha_{0,i}, \beta_{0,i})$ and then $\mu_i \sim \mathcal{N}(\mu_{0,i}, \frac{\sigma_i^2}{\kappa_{0,i}})$. Our regret bound is stated below.

**Theorem 2** (Variance-dependent regret bound for unknown variances). *Consider the above setting and let $\alpha_{0,i} \geq 1$ for all arms $i \in [K]$. Then for any $\delta > 0$, the Bayes regret of VarTS is bounded as:*

$$R_n \leq C\sqrt{n \log(1/\delta)} + \delta C\sqrt{nk/(2\pi)},$$

*where $C^2 = \sum_{i=1}^{K} \frac{\beta_{0,i}}{\alpha_{0,i}-1}\left(\frac{2}{\kappa_{0,i}} + \frac{0.5}{\kappa_{0,i}(\alpha_{0,i}-1)} + 5\log\left(1 + \frac{n}{\kappa_{0,i}}\right)\right)$ is a prior-dependent constant.*

**Proof of Theorem 2.** The difficulty lies in tightly bounding confidence intervals of random reward means with unknown reward variances, to obtain the right dependence on $\frac{\beta_{0,i}}{\alpha_{0,i}-1}$ and $\frac{1}{\kappa_{0,i}}$. This is algebraically challenging due complicated posterior updates of Gaussian-Gamma distributions in

Algorithm 2. To overcome these difficulties, we carefully condition random variables on each other together with their appropriate histories, and then combine them using Jensen's and Cauchy-Schwarz inequalities. The key lemmas along with the complete proof of Theorem 2 are in Appendix D.

**Dependence on all parameters of interest and prior.** For $\delta = 1/n$, the bound in Theorem 2 is $\tilde{O}(\sqrt{Cn})$. The dependence on $\sqrt{n}$ is the same as in Theorem 1. A closer examination of $C$ reveals many similarities with Theorem 1.

First, since $\sigma_i^{-2} \sim \text{Gam}(\alpha_{0,i}, \beta_{0,i})$, we know that $\sigma_i^2$ is sampled from an Inverse-Gamma distribution with the same parameters. The mean of this distribution is $\beta_{0,i}/(\alpha_{0,i} - 1)$. Hence, $\beta_{0,i}/(\alpha_{0,i} - 1)$ in Theorem 2 plays the role of $\sigma_i^2$ in Theorem 1 and represents the *effective reward variance*.

Second, $\kappa_{0,i}$ in the Gaussian-Gamma prior plays the role of $\sigma_i^2/\sigma_{0,i}^2$ in the known variance setting (Murphy, 2007). Therefore, as $\kappa_{0,i} \to \infty$, the bound in Theorem 2 should go to zero, similarly to Theorem 1. This is indeed the case and a very unique property of Bayes regret bounds, which is not captured by Honda & Takemura (2014); Zhou & Tian (2022).

Finally, we take $\alpha_{0,i}, \beta_{0,i} \to \infty$ while keeping $\beta_{0,i}/(\alpha_{0,i} - 1)$ fixed. Since the mean and variance of the Inverse-Gamma distribution are $\beta_{0,i}/(\alpha_{0,i} - 1)$ and $\beta_{0,i}^2/((\alpha_{0,i} - 1)^2(\alpha_{0,i} - 2))$, respectively, the mean of the variance prior is fixed while we narrow its width. In this case, we expect the bound in Theorem 2 to approach Theorem 1, which happens because $0.5/(\kappa_{0,i}(\alpha_{0,i} - 1))$ vanishes. After that, the bounds are similar up a multiplicative factor of 5.

**Existing frequentist regret bounds for variance-adaptive Thomson sampling.** Honda & Takemura (2014); Zhu & Tan (2020) proposed variance-adaptive Thompson sampling and bounded its regret. These works differ from us in three aspects. First, the algorithms of Honda & Takemura (2014); Zhu & Tan (2020) are designed for the frequentist setting. Specifically, they have a fixed sufficiently-wide prior, and enjoy a per-instance regret bound under this prior. As an example, the algorithm of Zhu & Tan (2020) for $\rho \to \infty$ (Remark 4) is essentially `VarTS` with $\mu_{0,i} = 0$, $\kappa_{0,i} = 0$, $\alpha_{0,i} = 0.5$, and $\beta_{0,i} = 0.5$. While per-instance regret bounds are strong, the algorithms of Honda & Takemura (2014); Zhu & Tan (2020) can perform poorly when priors are narrower and thus more informative. Truly Bayesian algorithm designs, as proposed in our work, can be analyzed for any informative prior. Second, the analyses of Honda & Takemura (2014); Zhu & Tan (2020) are frequentist. This means that they cannot justify the use of more informative priors and are essentially similar to those of frequentist upper confidence bound algorithms. As an example, in Remark 4 of Zhu & Tan (2020), the authors derive a $O\left(\sum_{i:\mu_i < \mu_{a^*}} \frac{1}{\Delta_i} \log n\right)$ regret bound, where $\Delta_i = \mu_{a^*} - \mu_i$ and $a^*$ is the arm with the highest mean reward $\mu_i$. This bound clearly does not depend on prior parameters, which we incorporate in our bounds. Specifically, our bound in Theorem 2 decreases with lower reward variances and more informative priors on them. Finally, the regret bounds of Honda & Takemura (2014); Zhu & Tan (2020) are asymptotic. We provide strong finite-time guarantees.

## 6 EXPERIMENTS

We also study the empirical performance of our proposed algorithms. Since `VarTS` does not assume that the reward variances are known, and thus is more realistic than Algorithm 1, we focus on `VarTS`. We conduct four experiments. First, we evaluate `VarTS` in a Bernoulli bandit, which is a standard bandit benchmark. Second, we experiment with beta reward distributions. Their support is $[0, 1]$, as in Bernoulli distributions, but their variances are not fully determined by their means. Since `VarTS` is designed for Gaussian bandits, the first two experiments also evaluate the robustness of `VarTS` to model misspecification. Third, we experiment with a Gaussian bandit. Finally, we vary the number of arms and observe how the performance of `VarTS` scales with problem size.

### 6.1 EXPERIMENTAL SETUP

All problem instances in our experiments are Bayesian bandits. The mean arm rewards are drawn from a prior distribution. In Gaussian bandits, `VarTS` is run with the true $(\boldsymbol{\mu}_0, \boldsymbol{\kappa}_0, \boldsymbol{\alpha}_0, \boldsymbol{\beta}_0)$. In other problems, the hyper-parameters of `VarTS` are set using the method of moments (Pearson, 1936) and samples from the prior. In particular, for a given Bayesian bandit, let $\bar{\mu}_i$ and $v_i$ be the estimated mean and variance of the mean reward of arm $i$ sampled from its prior, respectively. Moreover, let $\bar{\lambda}_i$ and $\nu_i$ be the estimated mean and variance of the precision of the reward distribution of arm $i$ sampled

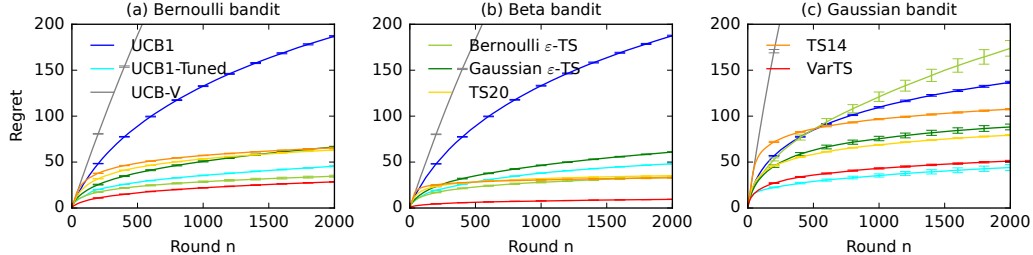

Figure 1: `VarTS` compared to 7 baselines. The plots share legends.

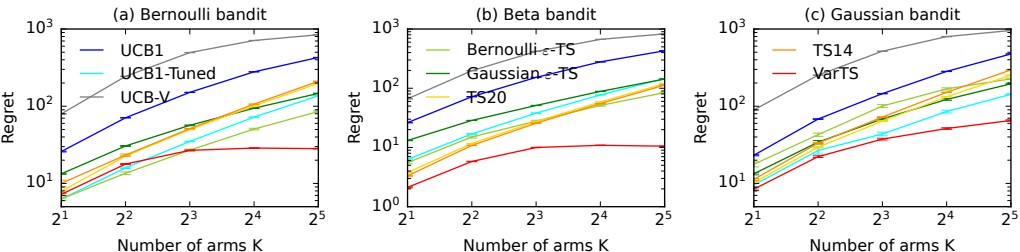

Figure 2: `VarTS` with 7 baselines as we vary the number of arms $K$. The plots share legends.

from its prior, respectively. Then, using these statistics, we estimate the unknown hyper-parameters of the prior as $\mu_{0,i} = \bar{\mu}_i$, $\beta_{0,i} = \bar{\lambda}_i / \nu_i$, $\alpha_{0,i} = \beta_{0,i}/\bar{\lambda}_i$, and $\kappa_{0,i} = \beta_{0,i}/(\alpha_{0,i} v_i)$.

We compare `VarTS` to several baselines. `UCB1` (Auer et al., 2002) is the most popular algorithm for stochastic $K$-armed bandits with $[0, 1]$ rewards. It does not adapt to reward variances and thus may be too conservative in our setting. We also consider variance-adaptive UCB algorithms: `UCB1-Tuned` (Auer et al., 2002) and `UCB-V` (Audibert et al., 2009b). `UCB1-Tuned` is a heuristic that performs well in practice. `UCB-V` uses empirical Bernstein confidence intervals and has theoretical guarantees. We implement both algorithms for $[0, 1]$ rewards. The next two baselines are Bernoulli and Gaussian TS (Agrawal & Goyal, 2013a). Bernoulli TS has a uniform $\text{Beta}(1, 1)$ prior. When the rewards $Y_{t,i}$ are not binary, we clip them to $[0, 1]$ and then apply Bernoulli rounding: the reward is replaced with 1 with probability $Y_{t,i}$ and 0 otherwise. Gaussian TS has a $\mathcal{N}(0, 1)$ prior and unit reward variances. We modify both algorithms to explore using the posterior sample $1/K$ fraction of time and the posterior mean otherwise, as proposed and analyzed in Jin et al. (2023). This significantly reduces the regret of Gaussian TS while the regret of Bernoulli TS remains comparable. We refer to these algorithms as $\varepsilon$-TS. The last two baselines are Thompson sampling with unknown reward variances (Honda & Takemura, 2014; Zhu & Tan, 2020). We implement Algorithm 1 in Honda & Takemura (2014) and call it `TS14`, and Algorithm 3 in Zhu & Tan (2020) for $\rho \to \infty$ and call it `TS20`. Note that `TS20` is `VarTS` where $\mu_{0,i} = 0$, $\kappa_{0,i} = 0$, $\alpha_{0,i} = 0.5$, and $\beta_{0,i} = 0.5$. The shortcoming of all TS baselines is that they are designed to have frequentist per-instance guarantees. Therefore, their priors are set too conservatively to compete with `VarTS`, which takes the true prior or its estimate as an input. In all simulations, the horizon is $n = 2\,000$ and they are averaged over $1\,000$ randomly initialized runs.

The baselines in our experiments were chosen to cover well existing variance-adaptive / not adaptive and Bayesian / frequentist bandit algorithms: `UCB1-Tuned`, `UCB-V`, `TS14`, and `TS20` adapt to unknown reward variances; while `UCB1` and $\varepsilon$-TS do not. $\varepsilon$-TS, `TS14`, and `TS20` are Bayesian algorithms; while `UCB1`, `UCB1-Tuned`, and `UCB-V` are frequentist algorithms.

## 6.2 BERNOULLI BANDIT

We start with a Bernoulli bandit with $K = 10$ arms. The mean reward of arm $i \in [K]$, $\mu_i$, is sampled i.i.d. from prior $\text{Beta}(i, K + 1 - i)$. Since $\mathbb{E}[\mu_i] = i/(K + 1)$ and $\text{std}[\mu_i] \approx 1/\sqrt{K + 1}$, higher prior means indicate higher $\mu_i$, although arm $K$ may not have the highest mean reward.

Our results are reported in Figure 1a. We observe that `VarTS` and Bernoulli $\varepsilon$-TS have the lowest regret. The latter is not surprising since Bernoulli $\varepsilon$-TS is designed for this problem class. The fact

that we match its performance is a testament to adapting to reward variances and using priors. Three out of four of the next best-performing algorithms (UCB1-Tuned, TS14, and TS20) adapt to reward variances but do not use informative priors. The frequentist algorithms with regret bounds (UCB1 and UCB-V) have the highest regret because they are too conservative.

### 6.3 BETA BANDIT

The bandit problem in the second experiment is a variant of Section 6.2 where the reward distribution of arm $i$ is $\text{Beta}(s\mu_i, s(1 - \mu_i))$ for $s = 10$. Roughly speaking, this means that the reward variance of arm $i$ is 10 lower than in Section 6.2. The rest of the setup is the same.

Our results are reported in Figure 1b. We observe only two differences from Figure 1a. First, VarTS outperforms Bernoulli $\varepsilon$-TS because it learns that the arms have 10 times lower reward variances than in Figure 1a. Therefore, it can be more aggressive in pulling the optimal arm. Second, both TS14 and TS20 outperform UCB1-Tuned, possibly due to more principled learning of reward variances.

### 6.4 GAUSSIAN BANDIT

The third experiment is with a Gaussian bandit where both the means and variances of rewards are sampled i.i.d. from a prior with parameters $\mu_{0,i} = i/(K + 1)$, $\kappa_{0,i} = K$, $\alpha_{0,i} = 4$, and $\beta_{0,i} = 1$. For this setting, $\mathbb{E}[\mu_i] = i/(K + 1)$ and $\text{std}[\mu_i] \approx 1/\sqrt{K + 1}$. Therefore, higher prior means indicate higher $\mu_i$, although arm $K$ may not have the highest mean reward. Since the average reward variance is 0.25, bandit algorithms for $[0, 1]$ rewards are expected to work well.

Our results are reported in Figure 1c. We observe that UCB1-Tuned has the lowest regret and VarTS performs similarly. This shows the practicality of our design, which is analyzable and comparable to a well-known heuristic without guarantees. All other algorithms have at least $50\%$ higher regret. As before, the frequentist algorithms with regret bounds (UCB1 and UCB-V) are overly conservative and among the worst performing baselines.

### 6.5 SCALABILITY

We vary the number of arms $K$ and observe how the performance of VarTS scales with problem size. This experiment is done in Bernoulli (Section 6.2), beta (Section 6.3), and Gaussian (Section 6.4) bandits. Our results in Figure 2 show that the gap between VarTS and the baselines increases with $K$. For $K = 32$ and Bernoulli bandit, VarTS has at least 3 times lower regret than any baseline. For $K = 32$ and beta bandit, VarTS has at least 5 times lower regret than any baseline. For $K = 32$ and Gaussian bandit, VarTS has at least 2 times lower regret than any baseline. These gains are driven by adaptation to reward variances and using priors, on both the mean and variance of the rewards.

## 7 CONCLUSIONS

We study the problem of learning to act in a multi-armed Bayesian bandit with Gaussian rewards and heterogeneous reward variances. As a first step, we present a Thompson sampling algorithm for the setting of known reward variances and bound its regret (Theorem 1). The bound scales as $\sqrt{n \log n} \sqrt{\sum_{i=1}^{K} \sigma_i^2 \log\left(1 + n\frac{\sigma_{0,i}^2}{\sigma_i^2}\right)}$. Therefore, it goes to zero as the reward variances $\sigma_i^2$ or the prior variances of the mean arm rewards $\sigma_{0,i}^2$ decrease. Our main contribution is VarTS, a variance-adaptive TS algorithm for Gaussian bandits with unknown heterogeneous reward variances. The algorithmic novelty lies in maintaining a joint Gaussian-Gamma posterior for the mean and variance of rewards of each arm. We prove a Bayes regret bound for VarTS (Theorem 2) that scales similarly to the known variance bound. More specifically, it is $\sqrt{n \log n} \sqrt{\sum_{i=1}^{K} \frac{\beta_{0,i}}{\alpha_{0,i} - 1} \log\left(1 + \frac{n}{\kappa_{0,i}}\right)}$, where $\frac{\beta_{0,i}}{\alpha_{0,i} - 1}$ is a proxy for the reward variance $\sigma_i^2$ and $\kappa_{0,i}^{-1}$ plays the role of $\sigma_{0,i}^2/\sigma_i^2$. Our bound captures the effect of the prior on learning reward variances and is the first such bound.

Potential future directions include extending our framework to infinite arms, which would require a different set of assumptions on the priors and reward distributions for tractable solutions. Another very practical and general direction would be incorporating context and changing reward variances over time, as in Kim et al. (2022); Zhao et al. (2022; 2023); Zhang et al. (2021).

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
