# OpenReview forum: "Only Pay for What Is Uncertain: Variance-Adaptive Thompson Sampling"
_ICLR.cc/2024/Conference — ICLR 2024 poster_

### Official Review · Reviewer_oe9F · 2023-10-23

**Soundness:** 3 good
**Presentation:** 3 good
**Contribution:** 3 good
**Rating:** 6
**Confidence:** 3

**Summary:**

The submission studied multi-armed bandits. The reward-generating process of each arm follows a different Gaussian-Gamma distribution. The submission proposed two Bayesian methods (Algorithms 1 and 2) and provided two prior-dependent regret bounds (Theorems 1 and 2). The proposed method (VarTS) is compared with seven existing methods in experiments.

**Strengths:**

The submission addresses the unknown reward variance. The regret is a finite-time analysis, and the bounds show how the reward variance affects the regret.

**Weaknesses:**

(a) Assuming Gaussian bandits would be impractical when facing realistic applications. As a complement, the analysis should discuss the cost of mis-modeling and show how to control this additional cost to obtain a meaningful regret bound. (In contrast, the experiments indeed consider distributions other than Gaussian.)

(b) The submission lacks a clarification that connects the novelty claimed in Contribution-(3) to the analysis in the appendix. This information would help evaluate and understand the technical contribution(s) of this submission.

(c) The variance-dependent bounds (Theorems 1 and 2) would be better justified if we could see curves of the proposed methods that vary according to the change of the unknown parameters in experiments.

**Questions:**

(d) The main paper categorizes the methods using Bayesian/frequentist approaches or the identical $\sigma$/distinct $\sigma_i$ settings. Could you please also reflect on these different baselines in the experiments, helping the reader to compare and contrast the differences between considering/not considering the prior information?

---

> ### Author Response · Authors · 2023-11-18
> **Rebuttal for Reviewer oe9F**
>
> Thank you for the detailed review. Our responses to your pointed-out weaknesses are below. We are happy to discuss any additional concerns that you may have.
>
> **W1: Theory beyond Gaussian bandits**
>
> A great question. One paper that analyzes the error of using Gaussian posteriors for other distributions is [Liu et al. (2022)](https://arxiv.org/pdf/2201.01902.pdf). Their results are quite pessimistic and lead to an additional $\varepsilon n$ term, where $\varepsilon$ is a measure of model misspecification. To make this term $O(\sqrt{n})$, an $O(1 / \sqrt{n})$ model misspecification would be needed. Since this is not practical, we opted for an empirical validation in Bernoulli and beta bandits.
>
> **W2: Novelty**
>
> The novelty is in analyzing a variance-adaptive algorithm. We derive the first Bayes regret bound for such an algorithm. The bound captures the benefit of informative priors and is finite time. None of the prior bounds have either of these properties. This is discussed in the last paragraph of Section 5.2. We discuss the technical novelty in Q3 in **Common response**.
>
> **W3: Plot the variance-dependent bound to validate it**
>
> ***In Appendix D of the updated paper, we plot the regret of VarTS, the regret bound in Theorem 2, and also frequentist regret bounds.*** We observe three trends:
>
> * The regret of VarTS decreases as the prior becomes more informative.
> * The regret bound of VarTS decreases as the prior becomes more informative.
> * Our regret bound becomes tighter than the frequentist bounds as the prior becomes more informative. This is because it depends on how informative the prior is. The expectation of the frequentist bounds does not depend on how informative the prior is.
>
> **Q1: Categorization of the baselines**
>
> We tried to cover all combinations of variance adaptive / not adaptive and Bayesian / frequentist bandit algorithms:
>
> * Adapts to unknown variance: UCB1-Tuned, UCB-V, TS20, TS14
> * Does not adapt to unknown variance: UCB1, Bernoulli TS, Gaussian TS
> * Bayesian algorithm: Bernoulli TS, Gaussian TS, TS20, TS14
> * Frequentist algorithm: UCB1, UCB1-Tuned, UCB-V

---

> ### Comment · Reviewer_oe9F · 2023-11-21
> **Thank you for your reply.**
>
> On W1: Based on the feedback, maybe adding a discussion similar to the following provides a general picture of the bandit problem. When one cannot control the misspecification, a frequentist approach is preferred, as an $\varepsilon n$ is a huge cost to pay. On the other hand, if one is confident with the underlying distribution (to be Gaussian), the Bayesian approach proposed in this submission would be the preferred one.
>
> On W2: Thank you for the feedback in Common Response. But I am looking forward to the discussion between you and reviewer Lfwz. (Although Lfwz decided to fix his/her score, there is no restriction to continue to defend your work.)

---

> ### Author Response · Authors · 2023-11-21
>
> Thanks for the follow-up comments.
>
> **Re. W1:** We totally agree with you on W1 and we will add this discussion as a remark in the updated version of our paper.
>
> **Re. W2:** We have further elaborated on our technical novelties and challenges addressed in the proof of Theorem 2 in Q3 in **Common response**, and also responded to Reviewer Lfwz on the follow-up concerns. Please go over them.
>
> Thanks again for your suggestions. Please let us know if we can clarify anything further.

---

### Official Review · Reviewer_tUNz · 2023-10-27

**Soundness:** 3 good
**Presentation:** 3 good
**Contribution:** 3 good
**Rating:** 8
**Confidence:** 3

**Summary:**

This paper addresses the challenge of optimizing bandit algorithms in the context of varying reward variances. Most bandit algorithms assume fixed reward variances or upper bounds on them, leading to suboptimal performance due to the inaccurate estimation of these variances. In this work, the authors lay the foundation for a Bayesian approach that incorporates prior knowledge about reward variances, resulting in lower regret and improved performance. They specifically focus on Gaussian bandits with unknown heterogeneous reward variances and develop a Thompson sampling algorithm with prior-dependent Bayes regret bounds.

**Strengths:**

They introduce a Thompson sampling algorithm for Gaussian bandits with known heterogeneous reward variances and provide regret bounds that decrease as reward variances decrease. This is a significant advancement in the context of bandit algorithms.

They propose a Thompson sampling algorithm (VarTS) for Gaussian bandits with unknown heterogeneous reward variances. VarTS maintains a joint Gaussian-Gamma posterior for mean rewards and precision, resulting in Bayes regret bounds that decrease with lower reward variances and more informative priors.

The paper thoroughly evaluates VarTS on various reward distributions, demonstrating its superiority over existing baselines. The results highlight the generality and robustness of the proposed algorithm.

The work distinguishes itself from prior research by providing strong finite-time regret guarantees, as opposed to asymptotic bounds.

They discuss the differences between their Bayesian approach and frequentist algorithms, highlighting the ability to leverage more informative priors in their design

**Weaknesses:**

The paper primarily focuses on Gaussian bandits with heterogeneous reward variances. While this is a significant step, the approach may not be directly applicable to other types of bandit problems with different reward distributions. The generalizability of the proposed method to various scenarios is not thoroughly explored.

Bayesian regret is often considered as easier than frequentist regret.

You may need to cite https://arxiv.org/abs/2006.06613 and https://arxiv.org/abs/2302.11182 as they analysed a Gaussian thompson sampling policy in a quite general setting which is related and relevant to the present paper.

I was thinking the Gaussian-Gamma prior was already used for the analysis of TS, but it seems that I am wrong since I cound not find back the ref.

**Questions:**

I am not convinced that this is a fundamental step in dealing with the problem of the general case of unknown variance in the case of non-Gaussian reward. Can you give more details on this ?

---

> ### Author Response · Authors · 2023-11-18
> **Rebuttal for Reviewer tUNz**
>
> Thank you for the detailed review. Our responses to your pointed-out weaknesses are below. We are happy to discuss any additional concerns that you may have.
>
> **W1: The approach is limited to Gaussian distributions with unknown variances**
>
> Please see Q2 in **Common response**.
>
> **W2: Bayes regret versus frequentist regret**
>
> Please see Q1 in **Common response**.
>
> **W3: Additional references**
>
> Thank you for the references. ***We included both [Perrault et al. (2020)](https://arxiv.org/abs/2006.06613) and [Perrault (2023)](https://arxiv.org/abs/2302.11182) in related works in the updated paper.*** However, note that these works focus on extending Thompson sampling to combinatorial bandits rather than variance adaptivity in our work. In a sense, this is an orthogonal structure.
>
> **Q1: Extension beyond Gaussian bandits**
>
> We discuss an extension beyond Gaussian bandits in Q2 in **Common response**. The general case is hard to handle in theory, and this is one limitation of Bayesian algorithms and their analyses. We discuss this in Q1 in **Common response**. The second best thing is an empirical validation and this is why we experiment with two types of sub-Gaussian noise in Section 6: Bernoulli and beta.

---

> > ### Comment · Reviewer_tUNz · 2023-11-22
> >
> > I acknowledge having read the rebuttal. I would like to thank the authors for their response to my questions. I will consider increasing my score.

---

> > > ### Author Response · Authors · 2023-11-22
> > > **Thank you**
> > >
> > > Dear Reviewer tUNz,
> > >
> > > Thank you for the response and considering increasing your score. We cited the two references you pointed us to. Specifically, your feedback on the frequentist regret, and generalization beyond Gaussian bandits was very helpful. Clarifying these points helped us to improve our paper a lot.
> > >
> > > Please let us know if we can clarify anything else.
> > >
> > > Sincerely,
> > >
> > > The authors

---

### Official Review · Reviewer_RHSZ · 2023-11-02

**Soundness:** 2 fair
**Presentation:** 2 fair
**Contribution:** 3 good
**Rating:** 6
**Confidence:** 4

**Summary:**

The authors investigate Gaussian bandits with heterogeneous reward variances, both known and unknown. For known variances, they introduce a Thompson sampling algorithm, achieving a Bayes regret bound that decreases as variances decrease. When variances are unknown, they present the VarTS algorithm, which employs a joint Gaussian-Gamma posterior for the mean and precision of all arm rewards. They establish a Bayes regret bound for VarTS that decreases with decreasing variances and stronger priors. Extensive experiments confirm VarTS's efficacy across different reward distributions.

**Strengths:**

1) This paper pioneers the study of Gaussian bandits with unknown heterogeneous reward variances and introduces the VarTS algorithm, leveraging a Gaussian-Gamma posterior for the task.
2) The authors provide Bayes regret bounds that captures the effect of the prior on learning reward variances for Gaussian bandits with both known and unknown heterogeneous reward variances. The regret analysis for the unknown variance scenario is novel.
3) Numerical experiments encompass not just Gaussian but also Bernoulli and beta distributions, demonstrating the generality and robustness of the proposed method.

**Weaknesses:**

1) As mentioned on page 5, the finite-time Bayes regret lower bound for Gaussian bandits remains unresolved, making the optimality of the Bayes regret bounds in Theorems 1 and 2 ambiguous. Furthermore, the current optimality discussion focuses on the order of $K, n$, but there should also be a discussion on its dependency on the prior and reward variances.
2) The algorithm design for Gaussian bandits with unknown heterogeneous rewards seems as standard as the typical TS algorithm.

**Questions:**

1) The detailed proof of Theorem 2 stands out as the main technical contribution in the paper. However, the current discussion on page 6 is somewhat succinct. A more in-depth exploration of the core ideas behind the proof would greatly benefit readers in grasping its implications.
2) Is it possible to derive finite-time prior-dependent regret bound for Bernoulli TS or Beta TS?

---

> ### Author Response · Authors · 2023-11-18
> **Rebuttal for Reviewer RHSZ**
>
> Thank you for the detailed review. Our responses to your pointed-out weaknesses are below. We are happy to discuss any additional concerns that you may have.
>
> **W1: Dependencies on prior and reward variances are not discussed**
>
> They should be. For Theorem 1, read starting from "Second, it increases with variances" on page 5. For Theorem 2, read starting from "Further, a closer examination of $C$ reveals many similarities:" on page 7. Please let us know if you would like to discuss anything else.
>
> **W2: Algorithm seems standard**
>
> Algorithm 2 is standard posterior sampling for Gaussian-Gamma distribution, based on well-known formulas in [Murphy (2007)](https://www.cs.ubc.ca/~murphyk/Papers/bayesGauss.pdf). The novelty is in analyzing a variance-adaptive algorithm. We derive the first Bayes regret bound for such an algorithm. The bound captures the benefit of informative priors and is finite time. None of the prior bounds have either of these properties. This is discussed in the last paragraph of Section 5.2.
>
> **Q1: Theorem 2 does not stand out as the main contribution**
>
> We will certainly elaborate on this in the main paper and include a detailed sketch. For the rebuttal, we discuss the technical novelty in Q3 in **Common response**.
>
> **Q2: Do Bernoulli or beta TS have finite-time prior-dependent regret bounds?**
>
> Bernoulli TS was analyzed in ranking problems by [Kveton et al. (2022)](https://proceedings.mlr.press/v151/kveton22a/kveton22a.pdf). Note that the Bernoulli distribution has a single parameter, which determines both its mean and variance. Beta TS is problematic because the beta distribution does not have a computationally tractable prior.

---

> > ### Comment · Reviewer_RHSZ · 2023-11-18
> >
> > Thank you for the response. To clarify my question about W1, I was asking whether the dependencies on prior and reward variances are tight, i.e., any lower bound result regarding them.

---

> > > ### Author Response · Authors · 2023-11-19
> > > **Further Clarification on W1 for Reviewer RHSZ**
> > >
> > > Thanks for clarifying W1 further. Our answer is below:
> > >
> > > **Known reward variances:** As mentioned in the last paragraph of Section 4.2, there are no finite-time $\Omega(\sqrt{K n})$ lower bounds for Bayes regret. The only known lower bound of this form is a $\Omega(\sqrt{K n})$ asymptotic lower bound of [Lattimore and Szepesvari. (2020)](https://tor-lattimore.com/downloads/book/book.pdf) (Theorem 35.1). We match it up to logarithmic factors when $\sigma_{0, i} = \sigma_i = 1$ for all $i \in [K]$. The lack of good lower bounds is a widely-recognized problem in Bayesian bandits. For instance, even the seminal works of Russo and Van Roy [(2014)](https://arxiv.org/abs/1301.2609) and [(2016)](https://www.jmlr.org/papers/volume17/14-087/14-087.pdf) that motivated all recent development do not provide any lower bound. Beyond a formal lower bound, there are other factors that indicate that the bound in Theorem 1 is of the right order:
> > >
> > > * **Increasingly informative prior:** As $\sigma_{0, i} \to 0$ for all $i \in [K]$, the bound in Theorem 1 goes to $0$. In this case, Thompson sampling knows the bandit instance before it interacts with it and therefore has no regret.
> > > * **Prior is not very informative:** When $\sigma_{0, i} \approx \sigma_i$ for all $i \in [K]$, the bound in Theorem 1 becomes $\tilde{O}\left(\sqrt{n \sum_{i = 1}^K \sigma_i^2} \right)$. This matches, up to logarithmic factors, gap-free regret bounds of frequentist bandit algorithms for heterogeneous reward variances.
> > > * **Doubling of reward variances:** In the frequentist setting, gap-dependent and gap-free regret bounds would increase by multiplicative factors of $2$ and $\sqrt{2}$, respectively, when the reward variances double. The gap-free regret bound in Theorem 1 would also increase by a multiplicative factor of $\sqrt{2}$, up to logarithmic factors.
> > >
> > > **Unknown reward variances:** When we discuss Theorem 2, in paragraph "Dependence on all parameters of interest and prior", we establish the following relation to Theorem 1:
> > >
> > > * $\beta_{0, i} / (\alpha_{0, i} - 1)$ in Theorem 2 is analogous to $\sigma_i^2$ in Theorem 1.
> > > * $\kappa_{0, i}$ in Theorem 2 is analogous to $\sigma_i^2 / \sigma_{0, i}^2$ in Theorem 1.
> > >
> > > Based on this, we conjecture that the dependencies on the variance parameters in Theorem 2 are of the right order, as in the known reward variance case.
> > >
> > > We hope that this answers your question. Please let us know if you have more questions.

---

### Official Review · Reviewer_u1uV · 2023-11-03

**Soundness:** 2 fair
**Presentation:** 3 good
**Contribution:** 2 fair
**Rating:** 6
**Confidence:** 4

**Summary:**

This paper studies the two-parameter Thompson Sampling with Bayesian regret. When the variance is known, the regret bound presented in this paper scales as $\sqrt{n\sum_{i=1}^{K}\sigma_i^2}\cdot \log n$, which is optimal up to a $\log n$ factor. In the case of unknown variance, the regret bound resembles that of the known variance scenario. Experimental findings indicate that the Thompson Sampling algorithm introduced in this study outperforms other baseline methods, including UCB and traditional Thompson Sampling.

**Strengths:**

See summary.

**Weaknesses:**

However, there are certain limitations to this paper:

1. The techniques used to prove Bayesian regret have been extensively explored in previous literature and are considerably more straightforward than those for frequentist regret. More crucially, Bayesian regret is a subset of frequentist regret. This is because the latter always implies Bayesian regret, but the reverse is not true.
2. The regret bound presented is not particularly stringent. It is a log n factor away from the optimal regret.
3. The paper omits some crucial related work such as:
    (1) Minimax policies for adversarial and stochastic bandits;
    (2) Mots: minimax optimal thompson sampling;
    (3) A minimax and asymptotically optimal algorithm for stochastic bandits;
    (4) Prior-free and prior-dependent regret bounds for thompson sampling
    (5) Thompson Sampling with less exploration is fast and optimal. It would be beneficial to see the baselines (2) (3) and (5) incorporated into the experiments.
4. The experimental framework is anchored in the Bayesian context. It would be enlightening to witness experiments in the general setting, i.e., no prior on the means and variance of arms.

**Questions:**

In Bayesian analysis, regret bounds are typically derived using the Upper Confidence Bound (UCB) method, as illustrated in works like "Prior-free and prior-dependent regret bounds for Thompson Sampling." However, this paper presents an alternative approach. It raises the question of whether it is possible to obtain the regret bound through a UCB method.

---

> ### Author Response · Authors · 2023-11-18
> **Rebuttal for Reviewer u1uV**
>
> Thank you for the detailed review. Our responses to your pointed-out weaknesses are below. We are happy to discuss any additional concerns that you may have.
>
> **W1: Bayes regret is weaker than frequentist regret**
>
> Please see Q1 and Q3 in **Common response**. In addition, note that we derive the first Bayes regret bound for variance-adaptive Thompson sampling. The bound reflects the benefit of informative priors and is finite time. None of the prior bounds have either of these properties. This is discussed in the last paragraph of Section 5.2.
>
> **W2: Presented bounds are not very stringent**
>
> Note that $\sqrt{\log n}$ grows much slower than $\sqrt{n}$. Therefore, it is customary to hide these factors in $\tilde O(\cdot)$ in bandit analyses. The elimination of the extra $\sqrt{\log n}$ is an interesting open question, and would be needed to close the gap between the regret upper and lower bounds.
>
> **W3: Paper omits some crucial related works**
>
> We respectfully disagree. The objective of our paper was to design, analyze, and empirically evaluate an algorithm that can adapt to unknown reward variances. Therefore, we included all relevant baselines to variance adaptation. That being said, we agree that we should use the latest improvements of Thompson sampling. We reviewed (2), (3), and (5). (5) compared their algorithm to (2) and (3), and showed that it is generally superior. ***Therefore, we implemented (5) for both Bernoulli and Gaussian Thompson sampling (Section 6 of the updated paper).*** This significantly reduces the regret of Gaussian TS while keeping the regret of Bernoulli TS comparable. Apart from this, all other conclusions in Section 6 remain the same.
>
> **W4: Bayesian framework in experiments**
>
> Bayesian algorithms are quite robust to prior and model misspecification. This can be seen in our experiments with Bernoulli and beta bandits, where VarTS (a Gaussian bandit algorithm) performs well. When the prior or model misspecification is major, Bayesian algorithms can fail miserably. We will comment on this in the paper.
>
> **Q1: Can we obtain a regret bound through a UCB-like analysis?**
>
> We follow the classic Thompson sampling analysis of [Russo and Van Roy (2014)](https://djrusso.github.io/docs/Learning_to_Optimize.pdf). The key idea is to bound the per-round regret conditioned on history by high-probability confidence intervals. Then we add them up and get an upper bound on the $n$-round regret. This style of an analysis is also common in linear bandits.

---

> > ### Comment · Reviewer_u1uV · 2023-11-21
> >
> > Thank you for your reply. I appreciate that you have addressed many of my concerns. However, there remains an issue regarding the related work section. While it is acceptable to incorporate recent studies in your experiments, it is also important to discuss other seminal papers in the field. This is particularly relevant since your paper considers the known variance setting—a scenario these previous works have significantly contributed to. Additionally, I recommend that you delineate between the modified Thompson Sampling, referred to as (5), and the original Thompson Sampling in your figures. This clarification is important because (5) represents a variation of the standard Thompson Sampling algorithm, and not the traditional version itself.

---

> ### Author Response · Authors · 2023-11-21
>
> Thanks for the follow-up comments.
>
> **Re. Related Works:** We completely agree with you that we should add a discussion of all the 5 (seminal) papers pointed out above in our related works as well. We were planning to add that in the updated submission, but now we will make sure to add these discussions in the related work section of our Openreview submission before November 22 and upload the draft again.
>
> **Re. (5):** Additionally, we will also update the plots to distinguish between (modified) $\epsilon$-TS (5) and standard (classic) TS. We will update them in the draft as well. Hope this will address all your concerns, please let us know if we can clarify anything further in the meantime.

---

> > ### Author Response · Authors · 2023-11-22
> > **Follow up**
> >
> > Dear Reviewer u1UV,
> >
> > As promised in the last response,
> >
> > * We added brief discussions of all your mentioned papers in Section 2. We highlighted the new text in blue and updated the draft on OpenReview. Please note all of these papers deal with classic $K$-armed bandits and they do not adapt to unknown variance. For convenience, we also added the same text below:
> >
> >   > Classic $K$-armed bandits have been studied for over three decades. Two widely used techniques for solving these problems are UCBs ([Auer et al., 2002](https://link.springer.com/article/10.1023/A:1013689704352); [Audibert & Bubeck, 2009](https://www.di.ens.fr/willow/pdfscurrent/COLT09a.pdf); [Menard & Garivier, 2017](https://proceedings.mlr.press/v76/m%C3%A9nard17a/m%C3%A9nard17a.pdf)) and Thompson sampling (TS) ([Agrawal & Goyal, 2012](https://arxiv.org/abs/1111.1797); [Bubeck & Liu, 2013](https://arxiv.org/pdf/1304.5758.pdf)). Recent works on TS matched the minimax optimal $\Theta(\sqrt{KT})$ rate in $K$-armed bandits ([Jin et al., 2021](https://proceedings.mlr.press/v139/jin21d/jin21d.pdf)). [Jin et al. (2023)](https://proceedings.mlr.press/v202/jin23b/jin23b.pdf) further designed a minimax and asymptotically optimal TS.
> >
> > * We also updated our experimental plots to distinguish the modified $\epsilon$-TS algorithm. The new plots can be found in the updated version of the paper in Section 6.
> >
> > Thanks again for all the suggestions, which greatly improved the paper. Please let us know if we can clarify anything further.

---

> > > ### Author Response · Authors · 2023-11-22
> > > **Thank you**
> > >
> > > Dear Reviewer u1UV,
> > >
> > > Thank you for the insightful discussion and increasing your score based on our responses. Your feedback, especially on baselines and related work, helped us to improve our paper a lot.
> > >
> > > Sincerely,
> > >
> > > The authors

---

### Official Review · Reviewer_Lfwz · 2023-11-11

**Soundness:** 3 good
**Presentation:** 3 good
**Contribution:** 2 fair
**Rating:** 5
**Confidence:** 3

**Summary:**

This work develops a new Thompson sampling algorithm and proves prior-dependent Bayes regret bounds for K-armed Gaussian bandits. It studies the problem in both settings where the reward variances are known and unknown. The algorithm can achieve lower regret when the
reward variances are low, which indicates the trade-off of the learner’s performance (regret) versus the prior parameters and reward
variances. The authors evaluate their new algorithms with numerical experiments comparing with other baselines.

**Strengths:**

1. The paper is well organized. The presentation is unambiguous.
2. The paper provides a new regret bound dependent on reward variances and informative priors.
3. The paper conducts numerical experiments to justify the main results.

**Weaknesses:**

1. The variance aware regret bound depends on the summation of the variance across all the arms $\sum_{i=1}^K \sigma_i^2$. This can
 reduce the dependency of the number of arms $K$ to a variance-dependent result. However, the standard method to deal with large $K$ is to use function approximation. I feel confused about the relationship of these two methods. Can the result in this work be generalized to the bandit problem with function approximation?
2. The Bayesian regret studied in this work is usually weaker than the frequentist regret.
3. Algorithm 2 is under the standard Thompson sampling framework, and the posterior sampling updates seem similar to previous works, for example, [Zhu & Tan., 2020].  Are there any novel techniques in the algorithm design, or do the new results come from the analysis?
4. There are some mistakes in the literature review. In the study of $d$-dimensional linear contextual bandits, they did not always keep the fixed variance across the arms. For example, in [Kim et al., 2022] and  [Zhao et al.,2023] mentioned in the paper, the stochastic noise at time step $k$, $\epsilon_k$, is a random variable dependent on $\mathcal F_k = \sigma(x_1,\epsilon_1,\ldots,x_{k-1},\epsilon_{k-1}, x_k)$ with $\mathbb{E}[\epsilon_k | \mathcal F_k] = 0$ and $\mathbb{E}[\epsilon_k^2 | \mathcal F_k] = \sigma_k^2$. They do not assume the variances are fixed across arms.
5. Some typos: should the bound in page 7 be $\tilde O(C\sqrt{n})$?

=============== Post Rebuttal ===================

Thanks for the response for the authors. After reading what the authors mentioned in the response, I still believe that there is no need to make such assumptions on the fixed variance across the arms. The notation $\sigma _ k ^ 2$ means the variance in round $k$, which can change when the selected arms are different (otherwise, the conditional expectation on $\mathcal F_k = \sigma(x_1,\epsilon_1,\ldots,x_{k-1},\epsilon_{k-1}, x_k)$ is meaningless.) I agree that assuming the fixed variance across the arms is a common practice. But for completeness, I suggest the authors mention the possibility of dealing with more general cases. I greatly appreciate the effort the authors made.  However, after careful consideration, I remain unconvinced about the paper's novelty, leading me to keep my scores.

**Questions:**

1. Is it possible to deal with distributions other than the Gaussian-Gamma prior distribution?

---

> ### Author Response · Authors · 2023-11-18
> **Rebuttal for Reviewer Lfwz**
>
> Thank you for the detailed review. Our responses to your pointed-out weaknesses are below. We are happy to discuss any additional concerns that you may have.
>
> **W1: Function approximations**
>
> Please note that the goal of this paper was to design, analyze, and empirically evaluate an algorithm that can adapt to unknown reward variances in the standard bandit setting. Our work does not focus on a large number of arms $K$. Therefore, no function approximation is used. Please see Q2 in **Common response** for a generalization beyond Gaussian bandits.
>
> **W2: Bayes regret is weaker than frequentist regret**
>
> Please see Q1 in **Common response**.
>
> **W3: Algorithmic or analysis novelty?**
>
> Algorithm 2 is standard posterior sampling for Gaussian-Gamma distribution, based on well-known formulas in [Murphy (2007)](https://www.cs.ubc.ca/~murphyk/Papers/bayesGauss.pdf). The novelty is in analyzing a variance-adaptive algorithm. We derive the first Bayes regret bound for such an algorithm. The bound captures the benefit of informative priors and is finite time. None of the prior bounds have either of these properties. This is discussed in the last paragraph of Section 5.2.
>
> **W4: Discussion of Kim et al. (2022) and Zhao et al. (2023)**
>
> There seems to be a misunderstanding of what we wrote. We say on page 2 that "$\sigma_s^2$ is an unknown reward variance **in round $s$**". We mean that the reward variance depends on round $s$. However, it is the same for all arms in round $s$. In our work, each arm has its own reward variance, and this one is the same for all rounds. We are happy to incorporate any additional changes if the reviewer has a different opinion and can further elaborate.
>
> **W5: Some typos**
>
> $\tilde O(\sqrt{Cn})$ should be $\tilde O(C \sqrt n)$. Thanks for pointing this out.
>
> **Q1: Other reward distributions than Gaussian-Gamma**
>
> Our algorithm design and analysis are structured enough to incorporate other models. We discuss this in Q2 in **Common response**.

---

> > ### Comment · Reviewer_Lfwz · 2023-11-20
> >
> > Thank you for your response. Please inform me if I have any misunderstanding, but I do not see any assumption that the variance is the same for all arms in round $s$, for example, in Kim et al. (2022). Could you provide any reference on such an assumption?

---

> > ### Comment · Reviewer_Lfwz · 2023-11-21
> >
> > While awaiting further clarification from the authors regarding the previous question on the variance assumption, upon reviewing the authors’ response on the novelty of the results, I have come to the realization that the algorithm seems rather conventional, and the analysis does not present any noteworthy novelty. The paper doesn't sufficiently discuss the novel techniques used in the analysis to establish the variance-dependent regret bounds.  The only discussion I can find is behind Theorem 2, which says “To overcome these difficulties, we carefully condition random variables on each other together with appropriate histories, and combine these using Jensen’s and Cauchy-Schwarz inequalities. ” To me, this is not novel. I have decided to maintain my current score and refrain from supporting its acceptance.

---

> ### Author Response · Authors · 2023-11-21
>
> Thanks for the follow-up comments.
>
> **Re. Further clarification on the discussion of Kim et al. (2022) and Zhao et al. (2023):** The main confusion could be in the notations used in their papers. Note that in these papers
>
> * K is the time horizon and $k \in [K]$ denotes the $k$-th time step in the time horizon K.
> * We on the other hand use $K$ to denote the total number of arms and $k \in [K]$ is the indexing used to denote the $k$-th arm among the $[K]$ arms.
>
> Now please read:
>
> * Page 2 last paragraph of [Kim et al. (2023)](https://arxiv.org/pdf/2111.03289.pdf): "Let K be the total number of rounds ..... till Equation (2.1)".
>
> * Section 2.1 of [Zhao et al. (2023)](https://proceedings.mlr.press/v195/zhao23a/zhao23a.pdf): "Linear bandits. The linear bandit problem has ..... till the last sentence of Page 2".
>
> Thus, in their notation $\sigma_k^2$ is the noise variance at time step $k$, which is same across all the arms, no matter which arm the algorithm selects in time step $k \in [K]$. To emphasize on our previous claim again,  these papers do not consider heterogeneous reward variance across arms -- unlike our setting -- where we assume distinct prior variance $\sigma_{0,i}$ as well as the distinct reward variance $\sigma_i$ for each distinct $i \in [K]$. Consequently, the nature of the results of these works are very different than ours, as they can not bring out the tradeoff between regret vs informative priors and variances (across the K arms), unlike our regret bounds in Theorem 1 and 2. Please let us know if you still have any confusion.
>
> **Re. Novelty:** We have further elaborated on our technical novelties and challenges addressed in the proof of Theorem 2 in Q3 in **Common response**, please go over it and also our detailed proof of Theorem 2 in the Appendix (Page 15-20). Hope this clarifies the novelties of our analysis of Theorem 2.
>
> * As promised, we will add a concise proof sketch of Theorem 2 in the updated version of our paper.
> * To emphasize again, as also clarified in Q3 in **Common response** above, our novelty not only lies in the proof of Theorem 2 but also in the implication of the final bound it reveals: We precisely obtained the ***First finite time regret bound for Bayesian bandits with unknown heterogeneous reward variances which captures the dependence of the regret on the degree of prior information and reward variances, unlike any other existing work.***
>
> Please let us know if we can clarify anything further.

---

### Author Response · Authors · 2023-11-18
**General Author Rebuttal: Common Response**

We would like to thank all reviewers for their feedback and appreciating our contributions. We will address three recurring points from the reviews here.

---
---
### Common Response ###
---
---

**Q1: Why a Bayesian analysis?**

Bayesian analyses are the only analyses in bandits that can capture the dependence on prior. In particular, as the prior becomes more informative, Bayes regret bounds go to zero, and so does the Bayes regret of Thompson sampling. Frequentist regret bounds do not have this behavior because they are proved for any bandit instance, which is unrelated to the prior used by the bandit algorithm. In fact, all frequentist regret bounds for Bayesian algorithms assume a sufficiently-wide prior, which is analogous to being uninformative. Taking an expectation of frequentist regret bounds over instances sampled from the prior does not yield the right dependence on the prior. ***In Appendix D of the updated paper, we plot the regret of VarTS, the regret bound in Theorem 2, and also frequentist regret bounds.*** As the prior becomes more informative, our bound becomes tighter than the frequentist bounds. This shows the benefit of Bayesian analyses.

Bayesian analyses have two shortcomings:

* They are on average over bandit instances sampled from the prior. This relates the bandit instances to the prior in the bandit algorithm and allows a prior-dependent analysis.
* To derive closed-form posteriors and use them in the analysis, modeling assumptions are needed. In our work, we assume Gaussian noise, which is less general than sub-Gaussian noise that is typically used in frequentist analyses. To show that VarTS works well beyond Gaussian noise, we experiment with two types of sub-Gaussian noise in Section 6: Bernoulli and beta.

**Q2: Beyond Gaussian distributions with unknown variances**

At a high level, our variance-adaptive algorithmic ideas and analyses can be applied to any exponential-family distribution with two parameters, where one represents the mean and the other represents the variance. One possibility is the gamma distribution. Its conjugate prior is listed in [A Compendium of Conjugate Priors](https://web.archive.org/web/20090529203101/http://www.people.cornell.edu/pages/df36/CONJINTRnew%20TEX.pdf) and [here](https://en.wikipedia.org/wiki/Conjugate_prior).

**Q3: Technical novelty in our analysis**

We derive the first finite-time regret bound for a variance-adaptive algorithm in Bayesian bandits with unknown heterogeneous reward variances. Our analyses show lower regret with lower reward variances (Theorem 1), even when they are initially unknown and learned (Theorem 2). Theorem 2 is our main result and it is novel for two reasons:

* First Bayes regret bound for unknown reward variances. ***The bound captures the benefit of informative priors, unlike any  existing bound for bandits with unknown reward variances.***
* First finite-time analysis of posterior sampling with unknown reward variances.

We discuss this in the last paragraph of Section 5.2. The proof of Theorem 2 is novel and goes significantly beyond prior works as follows:

* The key step in Bayes regret analyses is to bound the regret conditioned on history by a high-probability confidence interval width of the pulled arm. This quantity is typically $\sqrt{\sigma^2 \log(1 / \delta) / N}$, where $\sigma^2$ is the reward variance and $N$ is the number of pulls of the arm. The total regret corresponding to the arm after $m$ pulls is obtained by summing these for $N \in [m]$. We can obtain such a bound. However, $\sigma^2$ is replaced by the posterior sampled variance $\sigma_{t, i}^2$, which is random and changes between the rounds. Therefore, we cannot simply sum these up for $N \in [m]$.
* To eliminate $\sigma_{t, i}^2$, we show that it can be replaced by its mean, in expectation conditioned on history. The mean depends on $\alpha_{t, i}$, $\beta_{t, i}$, and $\kappa_{t, i}$ defined in VarTS. The main challenge in the rest of the proof is that these quantities are complex, involving ratios and squares of other random variables. In addition, $\alpha_{t, i}$ depends on random rewards. This is a significant departure from classic Bayes regret analyses, where the confidence intervals are $\sqrt{\sigma^2 \log(1 / \delta) / N}$ or can be bounded in this way, and the only random quantity is the number of pulls. Our analysis is based on a series of tight upper bounds, which maintain the expected dependence on prior quantities, as discussed after Theorem 2. The main proof is on pages 15-20 in Appendix.

We will add the above clarifications, including a sketch of the proof of Theorem 2, to the next version of the paper.

---

### Author Response · Authors · 2023-11-20
**Requesting follow up questions for further clarifications**

Dear reviewers,

We wanted to thank you for your insightful and detailed reviews. This allowed us to put together a good rebuttal and we submitted it 3 days ago. If you believe that we have not addressed your concerns, we would love to hear from you and discuss them.

Sincerely,

The authors

---

### Author Response · Authors · 2023-11-23
**Few Hours Until the Author-Reviewer Discussion Ends**

Dear Reviewers,

Thank you again for your time and insightful comments. Since we are only a few hours away from the end of the author-reviewer discussion phase, we wanted to check for the last time if we can clarify any remaining concerns. Please let us know and we would be happy to.

Sincerely,

The authors

---

### Meta-Review · Area_Chair_GBh9 · 2023-12-06

**Metareview:**

In this paper, the authors investigate Gaussian bandits with unknown heterogeneous reward variances. They introduce a Thompson sampling algorithm with prior-dependent Bayes regret bounds and conduct synthetic experiments to compare its performance against various existing frequentist approaches. The results demonstrate the superior performance of the proposed algorithm. Following author responses and reviewer discussions, the paper receives broad support from the reviewers. Thus I recommend acceptance.

**Justification For Why Not Higher Score:**

There are still minor concerns raised by one reviewer.

**Justification For Why Not Lower Score:**

The majority of the reviewers are in support!

---

### Decision · Program_Chairs · 2024-01-16

Accept (poster)